# TET1-TRPV4 Signaling Contributes to Bone Cancer Pain in Rats

**DOI:** 10.3390/brainsci13040644

**Published:** 2023-04-10

**Authors:** Zhen-Hua Xu, Zheng Niu, Yun Liu, Pei-Lin Liu, Xiao-Long Lin, Ling Zhang, Long Chen, Yu Song, Ren Sun, Hai-Long Zhang

**Affiliations:** 1Center for Translational Medicine, Affiliated Zhangjiagang Hospital of Soochow University, Zhangjiagang 215600, China; 2Department of Anesthesiology, Affiliated Zhangjiagang Hospital of Soochow University, Zhangjiagang 215600, China; 3Jiangsu Key Laboratory of Neuropsychiatric Diseases and Institute of Neuroscience, Soochow University, Suzhou 215123, China

**Keywords:** TET1, TRPV4, bone cancer pain, dorsal root ganglion, peripheral sensitization

## Abstract

Bone cancer pain (BCP) is excruciating for cancer patients, with limited clinical treatment options and significant side effects, due to the complex and unclear pathogenesis of bone cancer pain. Peripheral sensitization in dorsal root ganglion (DRG) neurons is a recognized cellular mechanism for bone cancer pain. The pathological mechanism of chronic pain is increasingly being affected by epigenetic mechanisms. In this study, we unbiasedly showed that the DNA hydroxymethylase ten-eleven translocation 1 (TET1) expression was significantly increased in the L4–6 DRG of BCP rats and ten-eleven translocation 2 (TET2) expression did not change significantly. Notably, TET1 inhibition by intrathecal injection of Bobcat339 (a TET1 inhibitor) effectively relieved mechanical hyperalgesia in BCP rats. Peripheral sensitization in chronic pain relies on the activation and overexpression of ion channels on neurons. Here, we demonstrated that TRPV4, one of the transient receptor potential ion channel family members, was significantly elevated in the L4–6 DRG of BCP rats. In addition, TRPV4 inhibition by intrathecal injection of HC067047 (a TRPV4 inhibitor) also significantly attenuated mechanical hyperalgesia in BCP rats. Interestingly, we found that TET1 inhibition downregulated TRPV4 expression in the L4–6 DRG of BCP rats. As a result, these findings suggested that TET1 may contribute to bone cancer pain by upregulating TRPV4 expression in the L4–6 DRG of BCP rats and that TET1 or TRPV4 may become therapeutic targets for bone cancer pain.

## 1. Introduction

There is severe pain associated with bone metastasis, which affects about 75% of patients with advanced breast cancer [1,2]. However, the exact cause of bone cancer pain is unknown, and the side effects of the available treatments are either intolerable or insufficient [3].

Peripheral sensitization is due to a sustained peripheral stimulation that makes primary sensory neurons undergo altered plasticity and become abnormally excitable [4,5]. Neurons undergo epigenetic changes to respond to harmful stimulation [6]. DNA demethylation is a classical epigenetic mechanism that participates in multiple physiological or pathological processes [7,8], such as chronic pain [9,10]. Demethylases regulate gene expression by catalyzing the oxidation of 5-methylcytosine (5-mc), producing 5-hydroxymethylcytosine (5-hmc), a derivative of 5-mc, which demethylates cytosine methylated at CpG islands [11,12]. The TET (Ten-eleven translocation) family of demethylases consists of three members: TET1, TET2, and TET3, with TET1 being known as the primary catalyst for 5-mc hydroxylation [13,14,15]. Some studies have demonstrated that intrathecal injection of exogenous TET1 causes mice to develop long-lasting nociceptive hypersensitivity [16,17]. TET1 expression is elevated in the spinal cord in inflammatory pain models, whereas TET1 inhibition effectively relieves inflammatory pain [18]. However, whether and how TET1 contributes to bone cancer pain in the rat DRG remains unclear.

TRPV4 is a member of the transient receptor potential (TRP) family of ion channels [19], previously named vanilloid receptor-related osmotically activated channel (VR-OAC), which functions in vivo in the transduction of osmotic and mechanical stimuli. In trpv4 null mice, TRPV4 was found to be necessary for the maintenance of systemic osmotic equilibrium, and for normal thresholds in response to noxious mechanical stimuli [20]. According to experimental studies, TRPV4 controls mechanical and thermal hyperalgesia in neuropathic pain [21,22]. TRPV4 inhibition partially reverses the mechanical pain after sustained compression of the dorsal root ganglion in rats [23]. Since the rapid proliferation of cancer cells severely squeezes the bone marrow cavity, which causes changes in cell surface pressure, and TRPV4 could sense changes in cell surface mechanical and osmotic pressure [24], we chose to study TRPV4 in the TRPV family. We used the online MethPrimer software to predict that there are two distinct CpG islands upstream of the transcription start site at the gene promoter of *TRPV4* in rats.

In this study, we hypothesized that TET1, a demethylase elevated in the DRG of rats with bone cancer pain, upregulates TRPV4 expression. To test the hypothesis, we investigated the expression and cellular localization of TET1 and TRPV4 in DRG and examined pain behaviors after drug treatment.

## 2. Experimental Procedures

### 2.1. Experimental Animals

We purchased female Sprague-Dawley (SD) rats weighing 180–220 g from Soochow University’s Laboratory Animal Center (Suzhou, Jiangsu, China). All rats were housed in a breeding room with a constant temperature of 23 °C, a 12/12 h light cycle, and plenty of food and water. The rats were required to adapt to the experimental environment for 2 days before the start of the experiment.

### 2.2. Culture of Tumor Cells

The Walker 256 tumor cell line was quickly defrosted in a water bath pan at 37 °C after being stored in liquid nitrogen. Following resuscitation, cell plates were washed with 1 × PBS buffer three times, and cell density was adjusted to 1 × 10^7^ cells/mL. Cells were cultured in high glucose (4.5 g/L) Dulbecco’s modified Eagle medium (Gibco, Thermo Fisher Scientific, Waltham, MA, USA), supplemented with 10% fetal bovine serum (Gibco, Thermo Fisher Scientific) and 1% penicillin/streptomycin (Sigma-Aldrich, St. Louis, MO, USA). These cells were then cultured in the presence of 5% CO_2_ at 37 °C. Then, using a syringe, 2.0 mL of this cell culture was injected into the peritoneal cavity of female SD juvenile rats weighing approximately 100 g. After feeding for one to two weeks, the rats were seen to have an abdominal bulge, which was cancerous ascites. Following local cleaning, ascites (5–10 mL) were aspirated from the rat’s abdominal cavity. Cells were treated with 1 × washing buffer, repeated three times, the cell density was adjusted to 1 × 10^7^/mL and aspirated at 10 µL and this cell culture was seeded into the bone marrow cavity of the left tibia of a rat, and the number of cells was 1 × 10^5^.

### 2.3. Establishment of Bone Cancer Pain Model

The bone cancer pain model was developed based on earlier research techniques [25,26,27]. In detail, after isoflurane inhalation anesthesia, the rat toe was lightly clamped with forceps, and no reaction allowed for modeling the surgery. The rat knee preps on the left hind limb were cleaned with scissors, then disinfected with iodophor cotton swabs. After exposing the tibial plateau, the musculofascial tissue was isolated by positioning the scalpel at this line through a roughly 0.5 cm incision. After locating the tibial tubercle with the dental probe and drilling a hole about 0.5 cm deep and gently cutting through the bone cortex, the needle was removed. Moreover, the adjusted Walker 256 tumor cell fluid was 10 µL (1 × 10^7^/mL) with microinjection needle aspiration, then, the needle was fed into the previous perforation, slowly pushing it into the bone marrow cavity of the left tibia of the rat, then medical glue was quickly dropped into the injection port after removing the needle, the surgical incision was then stitched, and the rat was placed on an electric blanket for rewarming. Sterilization guidelines were followed when performing surgical procedures and using related equipment.

### 2.4. Mechanical Pain Threshold Analysis

Pain behavior was recorded at 0 d, 3 d, 7 d, 14 d, and 21 d. The rats were put into a 20 cm × 20 cm × 25 cm organic glass cage with a metal screen underneath, then the rats were quiet for 30 min. The hind paw withdrawal mechanical threshold (PWT) in response to stimulation of Von Frey filaments (VFFs, UGOBasileSRL, Camerio, Italy) was determined as described previously [28]. An ascending series of calibrated VFFs (0.6, 1.0, 1.4, 2.0, 4.0, 6.0, 8.0, 10.0, 15.0, and 26.0 g) was used to stimulate the plantar surface of the left hind paw, and each VFF was held for about 1–2 s. When the rat responded by lifting or adding its foot, it was considered positive; otherwise, it was considered negative. First, a von Frey filament stimulus weighing 0.6 g was selected and administered using the “up and down” method until a rat responded positively [29]. Following that, corresponding values were recorded with a cutoff value of 26.0 g, with at least 5 min between trials, and an average of 6 consecutive measurements.

### 2.5. HE Staining

To confirm the effectiveness of the rat model for bone cancer pain by examining the invasion of tumor cells into the bone marrow cavity of the tibia, the tibia on the rat’s operated side was dissected, fixed with 10% PFA, and decalcified for 24 h in a 10% EDTA solution. Sections were then embedded in an embedding medium and sectioned to 10 µm. For Hematoxylin and Eosin (HE) staining, sections were rehydrated and stained with 0.1% Hematoxylin and 0.5% Eosin (Sigma-Aldrich) in sequence.

### 2.6. Western Blotting

Rats were given isoflurane inhalation anesthesia before being sacrificed at days 0, 3, 7, 14, and 21 after tumor cell injection. Following mechanical pain threshold detection, the L4–6 DRG were dissected on ice and collected. The collected DRG tissues were mixed with a protease inhibitor cocktail in a RIPA lysis buffer before being ultrasonically lysed to create tissue homogenates. After centrifuging the lysates at 12,000 g for 4 °C, the supernatant was collected, and the protein concentration was determined with a BCA protein assay kit (Thermo Scientific, Waltham, MA, USA). Equal amounts of protein were separated by 4–20% SDS-PAGE (ACE Biotechnology, Xiangtan, China) and transferred to PVDF membranes. Then, the membranes were blocked with 5% non-fat milk for 2 h at room temperature and probed with the primary antibodies anti-TET1 (Abcam, Boston, MA, USA, 1:1000), anti-TRPV4 (Santa Cruz, Dallas, TX, USA, 1:500), anti-β-tubulin (Proteintech, Rosemont, IL, USA, 1:20,000) at 4 °C overnight. The membrane was then incubated with peroxidase-conjugated secondary antibodies for 2 h at room temperature following TBST washing. The imaging system picked up the immunoreactive bands (BIO-RAD, Hercules, CA, USA).

### 2.7. Immunofluorescence

Rats from each group were fixed on the 14th day following the procedure while under isoflurane anesthesia. The heart was punctured at the apical part of the ascending aorta with an epidural needle, perfused with 500 mL of phosphate-buffered saline (PBS), and infused with 250 mL of 4% PFA. The L4–6 DRG was next dissected on ice and collected. The collected L4–6 DRG were placed into 4% PFA at 4 °C overnight, followed by sequential placement into 15% and 30% sucrose solutions for dehydration at 4 °C for 48 h. These L4–6 DRG samples were frozen in an OCT compound embedding medium and then sectioned into 15 µm thick sections by using a freezing microtome (Leica, Wetzlar, Germany). The slices were placed into 0.3% Triton X-100 at room temperature for half an hour. After that, sections were blocked with 5% BSA for 1 h at room temperature and washed with PBS. Additionally, the slides were incubated with the following primary antibodies at 4 °C overnight: anti-TET1 (Abcam, 1:200), anti-TRPV4 (Proteintech, 1:200), anti-NeuN (Merck Millipore, Burlington, MA, USA, 1:200), anti-GS (Genetex, Irvine, CA, USA, 1:200), anti-CGRP (Abcam, 1:200), anti-isolectin B4 (Sigma, 1:50). After washing with PBS three times, the secondary antibodies labeled Alexa Fluor 488 and 555 (Molecular Probes, Eugen, OR, USA, 1:800) were incubated at room temperature for 1 h. DAPI was used to stain the nuclei. Fluorescence microscope images were captured using AxioVision (Jena, Germany).

### 2.8. RT-qPCR

Rats were anesthetized with isoflurane inhalation and sacrificed at days 0, 3, 7, 14, and 21 following the injection of tumor cells and after drug management. The L4–6 DRG was then dissected on ice and collected. Trizon was used to extract total RNA from the L4–L6 DRG. Following the manufacturer’s instructions, an Omniscript RT kit 50 (Qiagen, Valencia, CA, USA) was used to synthesize cDNA from total RNA. The primer sequences of target genes, *TET1*, *TET2*, *TRPV4*, and the reference gene, *GAPDH*, were as follows:
5′-ATCCAGTGGGCAGGCATTTT-3′ (forward primer for *TET1*);5′-CACACTGGTTAGATGGAGGGG-3′ (reverse primer for *TET1*);5′-GGGATGTGGAAACGGCTACA-3′ (forward primer for *TET2*);5′-TCATCTTTTTCTGCATTTTTGCAC-3′ (reverse primer for *TET2*);5′-GCCACCCTACCCTTACCGTA-3′ (forward primer for *TRPV4*);5′-GGAAGGAGCCATCGACGAAGA-3′ (reverse primer for *TRPV4*);5′-GACAACTTTGGCATCGTGGA-3′ (forward primer for *GAPDH*);5′-ATGCAGGGATGATGTTCTGG-3′ (reverse primer for *GAPDH*).

### 2.9. Drug Administration

Under isoflurane inhalation anesthesia, the L4–5 spinous process space was punctured into the rats’ lower backs on the 8th day following the tumor cell inoculation. The rats were allowed to arch their lower backs with flat faces. The microsyringes were introduced into the needle slowly and vertically, with a tail flick motion that signaled the beginning of the injection of the drug after the syringe had been backfilled with cerebrospinal fluid. Bobcat339, a selective cytosine-based inhibitor of TET1, was purchased from MedChemExpress (HY-111558A). Bobcat339 (0.1 mg/kg, 20 µL) was dissolved in 0.9% normal saline and administered by intrathecal injection [30]. HC067047, a selective inhibitor of TRPV4, was purchased from MerkExpress (SML0143). HC067047 (0.1 mg/kg, 20 µL) was dissolved in 10% DMSO and administered by intrathecal injection [31]. We intrathecally injected either Bobcat339 or HC067047 for 7 consecutive days. On day 8 of BCP, the rats’ mechanical pain thresholds were assessed.

### 2.10. Statistical Analysis

Data were presented as mean ± SEM. Statistical analyses were conducted using GraphPad Prism 9 software. Comparisons between the two groups were analyzed with the two-sided *t*-test. Comparisons between three or more groups were analyzed with one-way ANOVA followed by Dunnett’s multiple comparisons tests. Two-way ANOVA was used to analyze the mechanical pain test data, and then Tukey’s multiple comparison tests were performed. Before analysis, the normality of all data was verified. The cutoff for statistical significance was *p* < 0.05.

## 3. Results

### 3.1. Walker256 Tumor Cell Inoculation Induced Mechanical Allodynia and Bone Destruction in Rats

After tumor cell inoculation, we observed bone destruction by dissecting the rat tibia from the surgical side to validate the bone cancer pain model. Additionally, we used Von Frey filaments to assess the paw withdrawal threshold (PWT) in rats. The results showed that the mechanical pain threshold in the BCP group decreased compared to the sham group, starting on day seven and declining until day fourteen, when it reached its lowest point, and then remained stable until day twenty-one (Figure 1A). Meanwhile, there was no change in mechanical pain threshold at each time point in the sham group compared with the naïve group. Tibial anatomy showed that the tibial tumors grew aggressively and induced bone destruction in the operated side of BCP rats compared with the sham group (Figure 1B). Additionally, Walker 256 tumor cell inoculation caused the trabecular bone structure to be severely destroyed, the bone cortex to erode, and tumor cells to fill the bone marrow cavity of the tibia. This information was provided by HE staining. This also further demonstrates the success of the bone cancer pain model (Figure 1C). The results indicated that Walker 256 tumor cell implanting led to mechanical allodynia and bone destruction.

### 3.2. The Expression of TET1 and TRPV4 Was Increased in L4–6 DRG of BCP Rats

To investigate the expression of TET1 and TRPV4 in L4–6 DRG of BCP rats, we performed qPCR and Western blotting at days 0, 3, 7, 14, and 21 after tumor cell inoculation. According to qPCR results, TET1 and TRPV4 mRNA levels gradually increased after the onset of bone cancer pain compared to those in the sham group (Figure 2A,C), but TET2 mRNA levels remained constant over time (Figure 2B). Western blotting showed that the TET1 expression increased from day 3 and persisted for 21 days after tumor cell inoculation with the progression of the bone tumor (Figure 2D), while the TET2 expression remained unchanged (Figure 2E) and the TRPV4 expression increased from day 7 and persisted for 21 days after tumor cell inoculation (Figure 2F). Notably, after tumor cell inoculation, TRPV4 expression increased from day 7 while TET1 expression started to rise on day 3 only (Figure 2D,F).

### 3.3. TET1 and TRPV4 Were Mainly Expressed in the L4–6 DRG CGRP and IB4 Neurons of BCP Rats

Using an immunofluorescence assay, we looked into the distribution of TET1 and TRPV4 in L4–6 DRG further. As shown in Figure 3, TET1 was mainly expressed in neurons (labeled by NeuN) but not satellite glial cells (labeled by GS, glutamine synthetase, a marker of satellite glial cells in DRG). TET1 was also co-stained with CGRP (calcitonin gene-related peptide, a marker of small and medium peptidergic neurons) and IB4 (isolectin B4, a marker of small and medium non-peptidergic neurons), and the results revealed that TET1 was mainly expressed in DRG CGRP or IB4 neurons (Figure 3). Additionally, we also investigated the distribution of TRPV4 in DRG and the results showed that TRPV4 was mainly colocalized with NeuN, CGRP, and IB4, but not with GS (Figure 4). Overall, the results showed that TET1 and TRPV4 were mainly distributed in DRG CGRP or IB4 neurons but not GS satellite glial cells (Figure 3 and Figure 4). TET1 is mainly localized in the cytoplasm and nucleus and could shuttle from the cytoplasm to the nucleus [32], while TET2 is mainly localized in the nucleus and is a nuclear protein [33]. We found an interesting difference between BCP rats and the sham group in that TET1 significantly shuttled from the cytoplasm to the nucleus, while TET2 remained in the nucleus without shuttle dynamic changes (Figure 5). The TET1 dynamic nucleus entry might be related to the role of TET1 as a DNA hydroxymethylase regulating gene transcription in BCP rats.

### 3.4. TET1 Inhibition Downregulated TRPV4 Expression in the L4–6 DRG of BCP Rats

To investigate whether TET1 regulates TRPV expression, we detected TRPV4 expression after intrathecal injection of the TET1 inhibitor Bobcat339 in the L4–6 DRG of BCP rats. We intrathecally injected Bobcat339 for 7 consecutive days on day 8 in BCP rats. The immunofluorescence results showed that TRPV4 positive cells decreased significantly after intrathecal injection of Bobcat339 in BCP rats, compared with the BCP + NS group (Figure 6A,C). Figure 6A–C shows that there were more TRPV4 positive cells in the L4–6 DRG of BCP rats than in the sham group, which is consistent with earlier qPCR and WB results (Figure 2C,F). Further proof came from the Western blotting results, which revealed that after intrathecal injection of Bobcat339 in BCP rats, as opposed to the BCP + NS group, TRPV4 expression was also decreased (Figure 6D,E). TRPV4 expression was increased in the L4–6 DRG of BCP rats, compared with the sham group (Figure 6D,E), which was consistent with the previous qPCR and WB results (Figure 2C,F). Interestingly, TET1 expression itself was also reduced after using the TET1 inhibitor (Figure 6F,G). Altogether, the results showed that TET1 inhibition downregulated TRPV4 expression in the L4–6 DRG of BCP rats.

### 3.5. TET1 or TRPV4 Inhibition Alleviated Mechanical Allodynia in BCP Rats

To verify whether TET1 or TRPV4 regulates bone cancer pain, we respectively injected the TET1 inhibitor Bobcat339 and TRPV4 inhibitor HC067047 intrathecally into BCP rats. We intrathecally injected either Bobcat339 or HC067047 for 7 consecutive days on day 8 in BCP rats. The results showed that TET1 or TRPV4 inhibition attenuated the mechanical allodynia in BCP rats, compared with the BCP + NS group (Figure 7). These results suggested that upregulated DRG TET1 and TRPV4 expression may be the cause of allodynia in rats with bone cancer pain and that allodynia may be reversed by pharmacologically inhibiting TET1 and TRPV4 activity.

## 4. Discussion

In this study, we showed that TET1 and TRPV4 expression was increased in L4–6 DRG after inoculation with Walker 256 tumor cells in rat tibia. Meanwhile, after inhibiting the TET1 activity, it decreased the expression of TRPV4 and relieved mechanical hyperalgesia in BCP rats. These findings suggested that TET1, or possibly TRPV4, could be used as a therapeutic target for bone cancer pain by upregulating TRPV4 expression in the L4–6 DRG of BCP rats.

DNA methylation is a classical epigenetic mechanism and participates in the pathogenesis of chronic pain [32,33,34]. There are three members of the TET protein family of hydroxymethylase: TET1, TET2, and TET3. TET1 is recognized as the main enzyme catalyzing 5-mc hydroxylation [13,14,15]. TET1 expression is elevated in the spinal cord in inflammatory pain models, whereas TET1 inhibition effectively relieves inflammatory pain [18]. However, whether TET1 is involved in bone cancer pain remains unclear. In this study, we successfully established a rat model of bone cancer pain and found higher expression of the demethylase TET1 in L4–6 DRG. In contrast, the expression of a different demethylase, TET2, did not appear to change significantly in the L4–6 DRG. This suggested that TET1 was specifically highly expressed in bone cancer pain compared to TET2. Using immunofluorescence localization, we discovered that TET1 was mainly expressed in L4–6 DRG neurons but not in satellite glial cells. Furthermore, we found that TET1 is significantly translocated to the nucleus in L4–6 DRG neurons from BCP rats. Therefore, we hypothesized that TET1 promotes the development of bone cancer pain by translocating to the nucleus to exert DNA demethylation. This is consistent with our finding that inhibition of TET1 led to a decrease in TRPV4 expression in BCP rats.

TRPV4, a member of the transient receptor potential ion channel receptor family [19], is crucial for many physiological processes, particularly the operation of the sensory system [21]. Different stimuli, such as osmotic pressure changes and mechanical, chemical, and thermal stimuli, could open the TRPV4 channels [21,22,23]. Additionally, TRPV4 has been implicated in mediating mechanical and hypotonic hyperalgesia in neuropathic pain model-induced pain [22]. In this study, we discovered that the L4–6 DRG of BCP rats had significantly higher TRPV4 expression compared to the sham group. Similarly to this, TRPV4 inhibition significantly reduced the abnormal pain experienced by rats with bone cancer pain. We used the online software MethPrimer to predict that there are two distinct CpG islands at the gene promoter of *TRPV4* in rats. Meanwhile, the demethylase TET1 regulates gene expression by catalyzing the demethylation of methylated cytosine at CpG islands. Therefore, we examined the expression changes of TRPV4 after intrathecal administration of TET1 inhibitors to rats with bone cancer pain. The results showed that TET1 inhibition downregulated TRPV4 expression in L4–6 DRG of BCP rats. As far as we know, this study was the first to show that TET1 inhibition downregulated TRPV4 expression and alleviated bone cancer pain in rats. However, the mechanism underlying TET1 upregulating TRPV4 under BCP conditions is not fully understood. Further studies are required to address whether TET1 upregulates TRPV4 expression by DNA demethylation at a CpG island of the *TRPV4* genes in BCP rats. Our study also had several limitations, including limited samples and a focus on only the BCP model. Further studies would use more samples, animal models, and advanced methods and examine bone cancer pain.

Even though both TET1 and TRPV4 were expressed in L4–6 DRG CGRP and IB4 neurons, and the inhibition of TET1 decreased the expression of TRPV4, it does not necessarily mean that the TET1 works through TRPV4 channels in the BCP rats. Although knockouts are the best way to confirm this, because TRPV4-KO rats have developmental disorders [34] and a long reproductive cycle, we used a TRPV4 inhibitor (HC067047) to investigate whether TET1 works through TRPV4 channels in the BCP rats. We found that the TRPV4 inhibitor (HC067047) significantly relieved bone cancer pain, which causes overexpression of TET1 and TRPV4, and that inhibition of TET1 reduces TRPV4 expression (Figure 6 and Figure 7). Therefore, we suggested that TET1 may contribute to bone cancer pain by upregulating TRPV4 expression in BCP rats. We found that TET1 expression was increased in BCP rats, and the TET1 inhibitor relieved bone cancer pain and decreased TRPV4 expression. Interestingly, with the use of the TET1 inhibitor for seven consecutive days, as the bone cancer pain was relieved, the expression of TET1 itself was followed by a decrease (Figure 6F,G). This suggested that the TET1 inhibitor may be a potential drug for the treatment of bone cancer pain.

## 5. Conclusions

In summary, the study suggests that TET1 contributes to bone cancer pain by upregulating TRPV4 expression in the L4–6 DRG of rats. TET1 or TRPV4 inhibition alleviates mechanical allodynia in BCP rats. These findings suggest that TET1 or TRPV4 may become effective therapeutic targets in cancer patients for pain relief.

## Figures and Tables

**Figure 1 brainsci-13-00644-f001:**
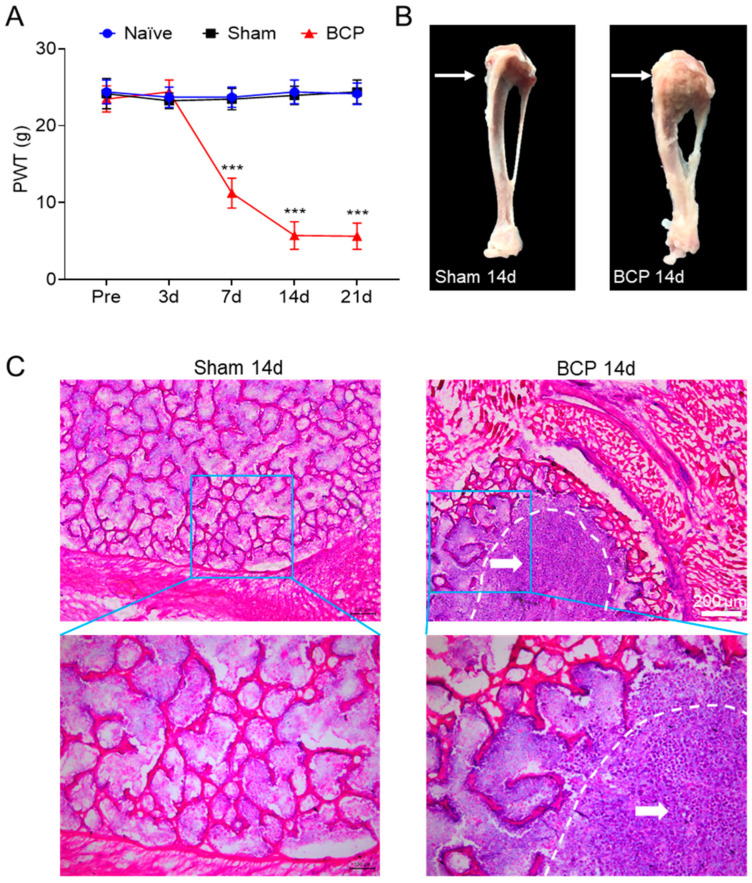
The bone cancer pain model was successfully established by injecting Walker 256 tumor cells into the tibia of rats. (**A**) The mechanical pain threshold was significantly reduced from day 7 to day 21 in rats after surgery compared with the naïve group and sham group (*** *p* < 0.001 vs. sham group, n = 8, two-way ANOVA followed by Tukey’s multiple comparisons test). Pre, preoperative. (**B**) We compared the tibia of the operated side in the BCP group and the sham group. The arrow points demonstrated that tumor expansion had destroyed the bone in the BCP rat. (**C**) The tibial bone in the sham rats was intact, and the arrangement of the trabecular bone structure was orderly, as demonstrated by HE staining. Red blood cells, lymphocytes, and macrophages are present in large numbers in the bone marrow cavity. However, the morphological changes of trabecular bone were lost, and the bone cortex was thinned and even destroyed in the BCP rats, who also had a large number of tumor cells filling the bone marrow cavity at the posterior end of the rat tibia. The region circled by the white dashed line represents the invasion of cancer cells.

**Figure 2 brainsci-13-00644-f002:**
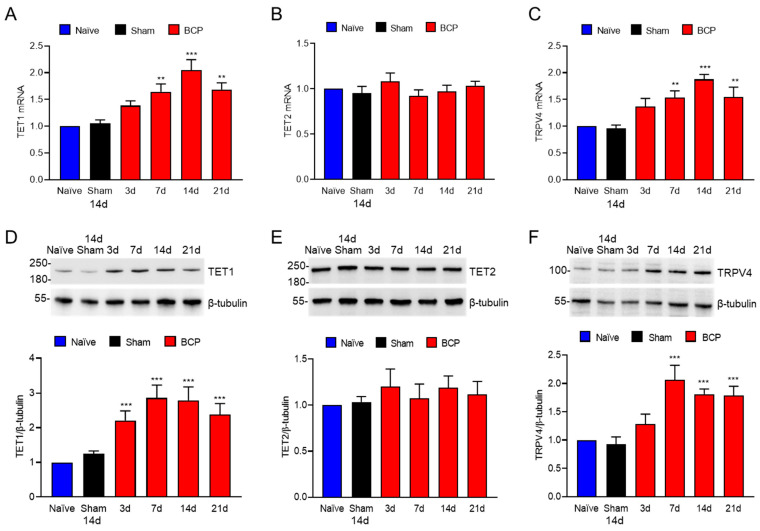
TET1 and TRPV4 were increased by qPCR and Western blotting at day 0, 3, 7, 14, and 21 in L4–6 DRG of BCP rats after tumor cell inoculation. (**A**) qPCR assay showed that TET1 mRNA increased with tumor progression (** *p* < 0.01, *** *p* < 0.001 vs. sham, n = 4, one-way ANOVA). (**B**) Compared with the sham group, TET2 mRNA has no significant difference at each time point. (**C**) qPCR assay showed that TRPV4 mRNA increased with tumor progression (** *p* < 0.01, *** *p* < 0.001 vs. sham, n = 4, one-way ANOVA). (**D**) Compared with the sham group, TET1 expression began to increase on the 3rd day after tumor cell inoculation and reached the highest level on the 7th day, and remained until the 21st day (*** *p* < 0.001 vs. sham, n = 4, one-way ANOVA). (**E**) TET2 expression was not significantly different from the sham group after tumor cell inoculation. (**F**) Compared with the sham group, TRPV4 expression began to increase on the 7th day after tumor cell inoculation (*** *p* < 0.001 vs. sham, n = 4, one-way ANOVA).

**Figure 3 brainsci-13-00644-f003:**
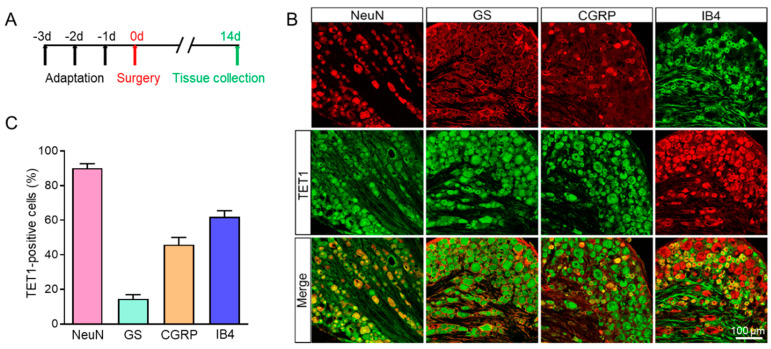
Immunofluorescence assay of TET1 in the L4–6 DRG of BCP rats. (**A**) The schematic diagram. (**B**) TET1-positive DRG cells (middle row) were mainly co-labeled as neurons (NeuN) but not with satellite glial cells (GS). TET1 was co-expressed in CGRP-positive and IB4-positive DRG neurons. (**C**) The proportion of TET1-positive cells in different cell types, n = 4.

**Figure 4 brainsci-13-00644-f004:**
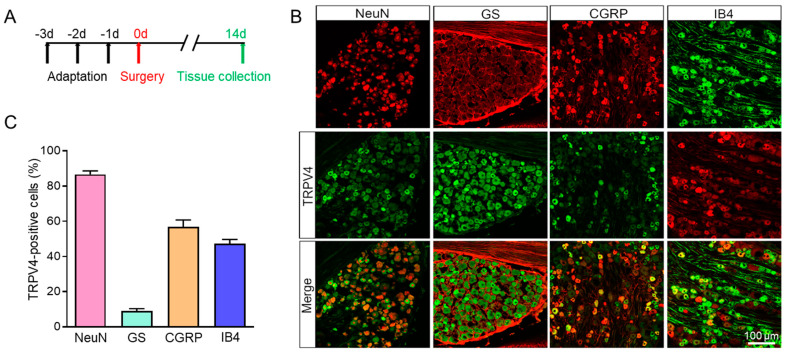
Immunofluorescence assay of TRPV4 in the L4–6 DRG of BCP rats. (**A**) The schematic diagram. (**B**) TRPV4-positive DRG cells (middle row) were mainly co-labeled as neurons (NeuN) but not with satellite glial cells (GS). TRPV4 was co-expressed in CGRP-positive and IB4-positive DRG neurons. (**C**) The proportion of TRPV4-positive cells in different cell types, n = 4.

**Figure 5 brainsci-13-00644-f005:**
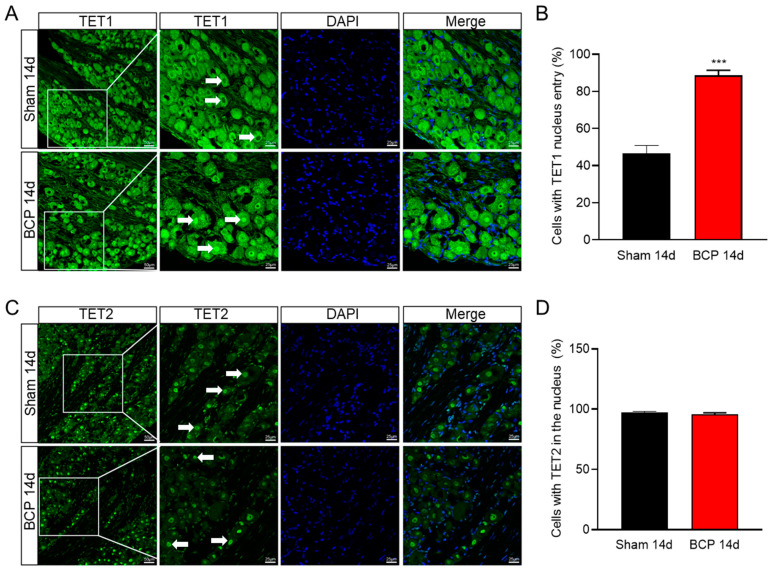
TET1 shuttled from the cytoplasm to the nucleus in BCP rats. (**A**,**B**) Compared with the sham group, the percentage of cells with TET1 nucleus entry was significantly increased in L4–6 DRG of BCP rats (*** *p* < 0.001 vs. sham, n = 4, two-tailed *t*-test). (**C**,**D**) No discernible difference in the proportion of cells with TET2 in the nucleus. Scale bar, 25 µm.

**Figure 6 brainsci-13-00644-f006:**
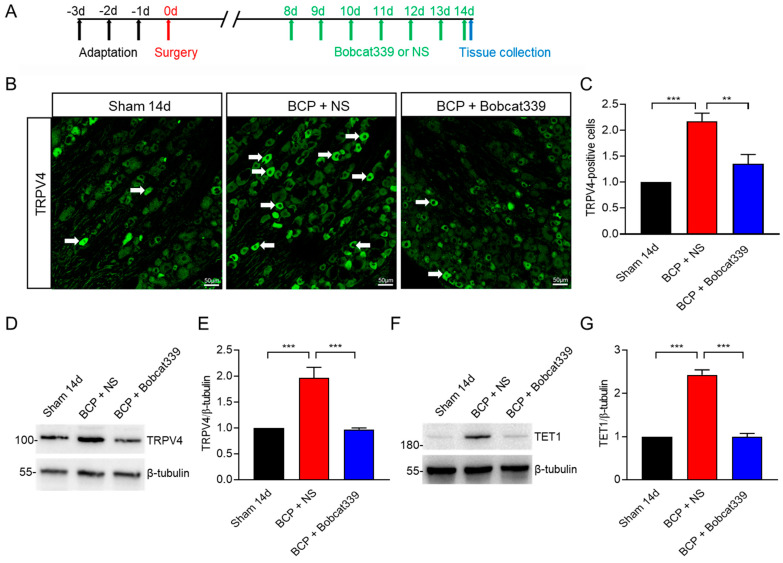
TET1 inhibitor Bob339 downregulated TRPV4 expression in L4−6 DRG of BCP rats. (**A**–**C**) The white arrows represent typical TRPV4−positive cells. After intrathecal injection of Bobcat339, there were significantly fewer TRPV4 positive cells in the L4−6 DRG of BCP rats than in the BCP + NS group (** *p* < 0.01, *** *p* < 0.001, n = 4, one−way ANOVA). Scale bar, 50 µm. (**D**,**E**) Compared with the BCP + NS, intrathecal injection of Bobcat339 significantly reduced TRPV4 expression in L4−6 DRG of BCP rats (*** *p* < 0.001, n = 4, one−way ANOVA). (**F**,**G**) Compared with the BCP + NS, intrathecal injection of Bobcat339 reduced TET1 expression in L4−6 DRG of BCP rats (*** *p* < 0.001, n = 4, one−way ANOVA).

**Figure 7 brainsci-13-00644-f007:**
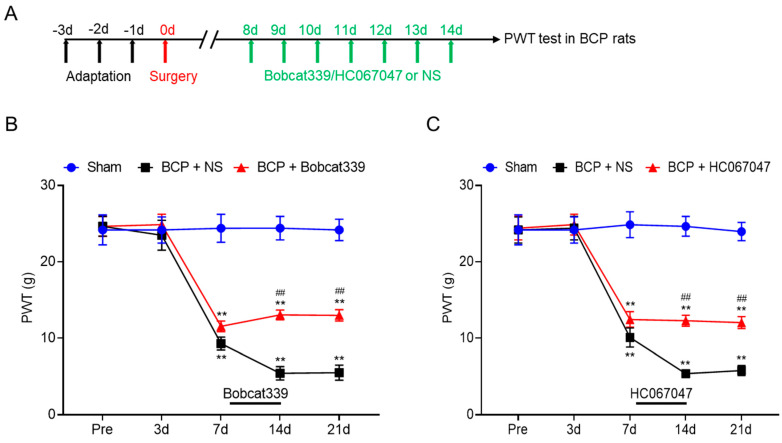
TET1 inhibitor Bob339 or TRPV4 inhibitor HC067047 alleviated mechanical allodynia in BCP rats. (**A**) The schematic diagram. (**B**) Bobcat339 was intrathecally injected on the 8th day after tumor cell inoculation for 7 consecutive days. Compared with the BCP + NS group, the allodynia of BCP rats was significantly partially reversed (** *p* < 0.01 vs. sham, ^##^
*p* < 0.01 vs. BCP + NS, n = 8, two−way ANOVA followed by Tukey’s multiple comparisons test). (**C**) HC067047 was intrathecally injected on the 8th day after tumor cell inoculation for 7 consecutive days. Compared with the BCP + NS group, the allodynia of BCP rats was significantly partially reversed (** *p* < 0.01 vs. sham, ^##^
*p* < 0.01 vs. BCP + NS, n = 8, two−way ANOVA followed by Tukey’s multiple comparisons test).

## Data Availability

The raw data supporting the conclusion of this article will be made available by the authors without undue reservation.

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
