# Peer review of "TET1-TRPV4 Signaling Contributes to Bone Cancer Pain in Rats"

_brainsci, 2023, doi:10.3390/brainsci13040644_

Round 1
Reviewer 1 Report
Comments and Suggestions for Authors
The manuscript presents clear evidence for the involvement of TET1-TRPV4 pathway in cancer-induced bone pain using a well-established animal model. Overall, the manuscript is well-written and presents the data in a clear and concise manner. However, there are some minor changes that could further improve the manuscript's clarity and impact.
As per the introduction, TET1 and TRPV4 are known to play a role in pain, and the study is focused on their expression in pain. However, it is not entirely clear why the authors chose to specifically study TRPV4. To address this, it may be beneficial for the authors to provide further explanation on their rationale for selecting TRPV4 over other TRPV channels or why they hypothesized a potential association between TET1 and TRPV4.
In Figure 1A, authors should demonstrate the progression of tumor cells in the tibia and establish a correlation between the rate of tumor growth and the degree of pain response measured by Von Frey testing.
In Figure 1C, it is challenging to distinguish between the marrow and tumor based on the H&E-stained images. To aid in differentiation between the two, it may be helpful to immunostain the sections with an antibody such as cytokeratin.
In Figure 2, the authors may need to provide an explanation for the lack of correlation between mRNA and protein expression of TET1 in DRG (as observed in Figures 2A and 2D). Notably, while the protein level of TET1 significantly increased at day 3, there was no change observed in its mRNA expression. Additionally, the authors mention three members of the TET family (TET1, TET2, and TET3); thus, it may be beneficial for the authors to clarify why TET3 expression was not explored in DRG of the cancer bone pain model in this study.
In Figure 3, there is room for improvement in the quality of the images presented. The images appear to show co-localization of TET1 and GS (yellow), indicating that TET1 is also expressed in satellite glial cells - which differs from the authors' conclusion. In addition, it would be beneficial for the authors to quantify the percentage of CGRP or IB4 neurons that show positive TET1 expression. This will help to better delineate the expression pattern of TET1 in these cells.
Figure 4. Same comment as Figure 3: it would be beneficial for the authors to quantify the percentage of CGRP or IB4 neurons that show positive TET1 expression.
For Figure 5, there is a need to improve the image quality, as it is not entirely clear from the images where TET1 and TET2 are localized in the cells. Additionally, it may be more informative if the results were presented as the percentage of cells with TET1 nucleus entry in Sham vs BCP, as opposed to fold change (as shown in panels B and D). This will help to provide a clearer understanding of the localization patterns of TET1 in the studied cells.
For Figure 6, there is a need for image quality improvement as it is challenging to distinguish the background from TRPV4-positive cells. Enhancing the image quality of these figures will help to improve the clarity of the presented data. This same issue is present in Figures 3, 4, and 5.
Regarding Figure 7, the authors could provide a demonstration of how the inhibitor affects TET1, specifically by indicating whether it affects the expression level or nucleus entry of TET1. Moreover, it would be helpful to include information in the manuscript to confirm that the tumor growth in the animal was not affected by the inhibitor injection, despite the inhibitor being delivered locally. This will help to ensure that the study's findings are not confounded by any effects on tumor growth caused by the inhibitor injection.
Reviewer 2 Report
Comments and Suggestions for Authors
The paper entitled “TET1-TRPV4 Signaling Contributes to Bone Cancer Pain in Rats” provides the insights about possible factors that may play roles in bone cancer pain. The authors found that expressions of both TET1 and TRPV4 were upregulated in the BCP rat model. Also, they proposed that the TET1 may contribute to bone cancer pain via regulating the expression of TRPV4. Overall, this paper suggests a possible signaling pathway that may be related to bone cancer pain. However, in order to support this conclusion, more evidence need to be provided.
First, the introduction about the TRPV4 is not accurate and sufficient. Also, the methods section needs to be improved. For example, in the 2.2 Culture of tumor cells section, more details need to be provided such as culture medium, CO2 levels, detailed washing procedures and so on. Also, the HE staining methods are not well described. It would be good if the authors can provide sources of essential materials. Experiments wise, for figure 2, the authors compared RNA and protein levels of TET1 and TRPV4 on day 0, 3, 7, 14, and 21 in L4-6 DRG of BCP rats after tumor cell inoculation with sham operation. However, it was not stated how long they waited after the operation for the sham group. Even though both TET1 and TRPV4 were expressed in L4-6 DRG CGRP and IB4 neurons, and the inhibition of TET1 decreased the expression of TRPV4, it does not necessarily mean that the TET1 works through TRPV4 channels in the BCP rats. In order to support the hypothesis, the TRPV4-KO rats are more direct experiments to do. Besides, please fix some typos and some confusing sentences.
Reviewer 3 Report
Comments and Suggestions for Authors
TET1-TRPV4 Signaling Contributes to Bone Cancer Pain in 2 Rats
Authors have explained the association of TET1, TRPV4 and BCP in the present manuscript. There are some concerns and suggestions for the authors.
1. How was the sample size calculated
2. How many rats were housed in one cage
3. Why did authors select Sprague -Dawley rats for the BCP model
4. The magnification at which images were taken
5. Authors are suggested to explain a bit more about TET1 and DRG and their association with BCP in the introduction section
6. Authors should add experimental timeline
Round 2
Reviewer 2 Report
Comments and Suggestions for Authors
The author addressed my comments in the revised version.